# E2F1 Promotes Progression of Bladder Cancer by Modulating RAD54L Involved in Homologous Recombination Repair

**DOI:** 10.3390/ijms21239025

**Published:** 2020-11-27

**Authors:** Jeong-Yeon Mun, Seung-Woo Baek, Won Young Park, Won-Tae Kim, Seon-Kyu Kim, Yun-Gil Roh, Mi-So Jeong, Gi-Eun Yang, Jong-Ho Lee, Jin Woong Chung, Yung Hyun Choi, In-Sun Chu, Sun-Hee Leem

**Affiliations:** 1Department of Biological Sciences, Dong-A University, Busan 49315, Korea; jeongyeonmun90@gmail.com (J.-Y.M.); kimwt33@nate.com (W.-T.K.); royunkil@gmail.com (Y.-G.R.); dkwl523@hanmail.net (M.-S.J.); boos9063@naver.com (G.-E.Y.); Topljh19@dau.ac.kr (J.-H.L.); jwchung@dau.ac.kr (J.W.C.); 2Department of Bioinformatics, KRIBB School of Bioscience, Korea University of Science and Technology, Daejeon 34113, Korea; baek@kribb.re.kr (S.-W.B.); seonkyu@kribb.re.kr (S.-K.K.); chu@kribb.re.kr (I.-S.C.); 3Genome Editing Research Center, Korea Research Institute of Bioscience and Biotechnology (KRIBB), Daejeon 34141, Korea; 4Department of Pathology, Pusan National University Hospital, Busan 49241, Korea; arkray-hyde@hanmail.net; 5Research Center, Dongnam Institute of Radiological & Medical Science (DIRAMS), Busan 46033, Korea; 6Personalized Genomic Medicine Research Center, Korea Research Institute of Bioscience and Biotechnology (KRIBB), Daejeon 34141, Korea; 7Department of Health Sciences, Dong-A University, Busan 49315, Korea; 8Department of Biochemistry, College of Korean Medicine, Dong-eui University, Busan 47227, Korea; choiyh@deu.ac.kr; 9Anti-Aging Research Center, Dong-eui University, Busan 47340, Korea

**Keywords:** bladder cancer, homologous recombination repair, E2F1, RAD54L, prognostic biomarkers

## Abstract

DNA repair defects are important factors in cancer development. High DNA repair activity can affect cancer progression and chemoresistance. DNA double-strand breaks in cancer cells caused by anticancer agents can be restored by non-homologous end joining (NHEJ) and homologous recombination repair (HRR). Our previous study has identified E2F1 as a key gene in bladder cancer progression. In this study, DNA repair genes related to E2F1 were analyzed, and RAD54L involved in HRR was identified. In gene expression analysis of bladder cancer patients, the survival of patients with high RAD54L expression was shorter with cancer progression than in patients with low RAD54L expression. This study also revealed that E2F1 directly binds to the promoter region of RAD54L and regulates the transcription of RAD54L related to the HRR pathway. This study also confirmed that DNA breaks are repaired by RAD54L induced by E2F1 in bladder cancer cells treated with MMC. In summary, RAD54L was identified as a new target directly regulated by E2F1. Our results suggest that, E2F1 and RAD54L could be used as diagnostic markers for bladder cancer progression and represent potential therapeutic targets.

## 1. Introduction

In 2019, about 80,000 adult men in the United States were diagnosed with bladder cancer (BC), making BC the fourth most common cancer [1]. The diagnosis of bladder cancer is four-fold higher in men than in women. Nearly 90% of patients diagnosed with BC are 55 years old or more. The average age at the time of diagnosis is 73 years. The incidence of BC increases with age [1]. During the initial diagnosis, about 75% of people with BC are diagnosed with non-muscle invasive bladder cancer (NMIBC) and the remaining 25% are diagnosed with muscle invasive bladder cancer (MIBC) or metastatic BC [2]. Patients with NMIBC are generally treated with transurethral resection (TUR) and/or intravesical instillation into the bladder. Their 5-year prognosis is 80% or more, which is relatively good [3]. However, the 5-year recurrence rate for NMIBC ranges from 31% to 78%, and the progression rate of NMIBC to MIBC varies from 0.8% to 45% [4]. In addition, those with progression to MIBC have poor prognosis because only 50% of patients who have undergone acute cystectomy can survive after five years [5].

To prevent such progression and recurrence, radiation and anticancer drugs can be used in addition to surgical treatment. However, many studies have reported cancer progression due to resistance of cancer cells to these additional treatments [6]. Several anticancer agents can cause DNA double-strand breaks (DSBs) that are considered fatal DNA lesions to cells [7]. DSBs in the genome generated by anticancer drugs can be repaired via non-homologous end joining (NHEJ) and homologous recombination repair (HRR) pathways [8]. In the NHEJ mechanism for DSB repair, the insertions or deletions at breakpoints can increase genomic instability and heterogeneity of cancer cells [9]. The HRR mechanism is important for maintaining genomic stability via error-free DNA DSB recovery. HRR is used by cancer cells to enhance chemical resistance [10].

Mutations of various genes involved in HRR (*BRCA1, BRCA2, RAD51, RAD54, XRCC2* or *XRCC3*) can lead to cancer development. They have been found in hereditary breast cancer, uterine cancer, colon cancer, melanoma, ovarian cancer, and pancreatic cancer [11,12]. Overexpression of *RAD51* in pancreatic and prostate cancers is correlated with decreased patient survival [13]. MCF cells with *XRCC3* overexpression are associated with higher tumorigenic capacity in vivo [14]. According to recent studies, inactivated HRR mechanisms have significant effects on cancer cell survival. They can increase the sensitivity of cancer cells to anticancer agents such as crosslinking agents (e.g., mitomycin C) [15,16]. Therefore, it is possible to develop a cancer therapeutic target by focusing on DNA repair mechanisms in response to anticancer therapy and resistant mechanisms.

To elucidate bladder cancer progression, our previous study has analyzed gene expression profiles of bladder cancer patients and identified a transcription factor *E2F1*, a key gene for cancer progression [17]. *E2F1* is known as a transcription regulator of various processes such as cell cycle, apoptosis, proliferation, development, and differentiation [18]. *E2F1* is also an important transcription factor in DNA repair. It can be induced by various DNA damaging stimuli, including ionizing radiation, UV radiation, and numerous chemotherapeutic agents [19,20]. Therefore, this study focused on the expression of DNA repair-related genes related to *E2F1* transcription factor associated with cancer progression. The results showed that RAD54L gene had the strongest correlation with *E2F1*. *RAD54L* plays an important role in HRR of DSBs. In this study, we investigated the correlation between *E2F1* and *RAD54L* and the effect of such correlation on cancer progression. *RAD54L* was identified as a new *E2F1* target. Results revealed that *E2F1* was involved in the HRR mechanism by regulating the expression of *RAD54L.* In addition, a group of bladder cancer patients with high expression of *RAD54L* had worse prognosis for cancer progression, suggesting that *RAD54L* in addition to *E2F1* is a new biomarker for cancer progression.

## 2. Results

### 2.1. DNA Repair Gene RAD54L Is Strongly Correlated with E2F1 in Gene Expression Profiles of Bladder Cancer Patients

Our previous study [17] has shown that E2F1 is a potential prognostic marker that can predict the progression from superficial bladder cancer to invasive bladder cancer. Transcription factor E2F1 is involved in cell cycle, apoptosis, and proliferation. Increased expression of E2F1 has been associated with DNA damage and tumor development [19]. High DNA repair activity is associated with cancer progression or recurrence. Therefore, gene expression profiling data of 165 bladder cancer patients (GSE13507) were used to identify genes highly correlated with the expression of transcription factor E2F1. RAD54L related to DNA repair showed the highest correlation (*R* = 0.901) with E2F1 expression. Gene-gene network analysis showed the significant role of E2F1-correlated genes (e.g., *E2F1, RAD54L, CDCA5, TRIP13, POLQ, EXO1, HMGB2, BLM, UBE2T, BRCA1, RECQL4,* and *RAD51*) in DNA repair.

These genes are associated with the repair of DSBs and re-combinational repair (Figure 1A, Appendix A). Using gene expression data of 165 bladder cancer patients in GSE13507, 1284 genes correlated with RAD54L (Pearson correlation test: *P* < 0.001, *r* > |0.4|) were selected. Based on expression profiles of these 1284 genes, a hierarchical cluster analysis was performed by dividing 165 bladder cancer patients into high RAD54L clusters (high RAD54L expression, red) and low RAD54L clusters (low RAD54L expression, blue) (Figure 1A). It has been reported that DNA repair mechanisms are associated with cancer incidence and prognosis [21,22,23]. Thus, we performed cancer-specific survival (CSS) and progression-free survival (PFS) analysis to determine the prognostic value with a log-rank test. The group with high expression levels of RAD54L was found to have a significantly lower CSS (*P* < 0.001) and PFS (*P* < 0.001) (Figure 1B). As a result, we assessed the group with relatively high levels of RAD54L expression showing poor prognosis.

### 2.2. E2F1 Directly Regulates HRR-Related RAD54L Expression

Abnormal expression and gene amplification of E2F1 has been observed in many types of human cancer [24]. Thus, the expression levels of E2F1 in five human bladder cancer cell lines were analyzed. The following experiments were performed using EJ cell line selected from cell lines with high E2F1 expression (RT4, T24 and EJ) and 5637 cell line selected from cell lines (KU7 and 5637) with relatively low E2F1 expression (Figure 2A). To investigate whether E2F1 affected the expression of RAD54L, 5637 cells were transiently transfected with an E2F1 overexpression vector (pcDNA-E2F1; pE2F1) or empty pcDNA as a negative control. The expression of E2F1 and RAD54L was evaluated by RT-PCR and western blot analysis (Figure 2B). Results showed that E2F1overexpression significantly increased the expression of RAD54L (Figure 2B). To determine the effect of E2F1 downregulation on RAD54L expression, stable E2F1-knockdown cells (shE2F1) were constructed using EJ cells. The levels of both RAD54L mRNA and protein were reduced in EJ cells with E2F1 knockdown (shE2F1) more than in control cells (shNTS) (Figure 2C). These results indicate that the expression of RAD54L can be increased or decreased depending on the levels of E2F1 in bladder cancer cells, suggesting that RAD54L is regulated by transcription factor E2F1.

Next, we analyzed the sequence of RAD54L to determine whether or not the transcription factor E2F1 binds to the promoter region of RAD54L and directly affects its transcriptional activity. We identified two putative E2F1 binding sites (TTCCCGC) in the promoter region of RAD54L and constructed a luciferase vector with the RAD54L promoter (Figure 3A). The RAD54L promoter vector (pro-RAD54L) was constructed by inserting a promoter region of approximately 1.4 kb (−1191/+210) containing the identified E2F1-binding sequences #1 (−767/−760) and #2 (−155/−148) (Figure 3A). The expression of the reporter vector was induced in a dose-dependent manner by the transient transfection of 5637 and EJ cells with this vector (Figure 3B).

Chromatin immunoprecipitation (ChIP) was used to examine the binding affinity of E2F1 for the promoter region of RAD54L to reveal whether E2F1 directly regulated RAD54L. Appendix A lists qRT-PCR primer sets used to confirm the binding at two E2F1-binding sites (#1, #2) and non-target site (NTS) on RAD54L. Control pcDNA or pE2F1 was transfected into 5637 cells. ChIP assay was then performed using control IgG and E2F1 antibodies. Results showed no difference in #2 and non-target sequence (NTS) regions. However, when #1 region was used, PCR fragment was significantly amplified in pE2F1-transfected cells (Figure 3C). These results showed that E2F1 directly binds to #1 region of the RAD54L promoter to regulate RAD54L transcription.

### 2.3. E2F1 Is Involved in HRR Pathway via Regulation of RAD54L

Mitomycin C (MMC) is a potent anticancer drug acting as a bifunctional alkylating agent that can lead to intrastrand and interstrand DNA cross-links, the most toxic complexes generating DSBs [25]. HR plays a critical role in the repair of DSBs. It uses the homologous sequence of a sister chromatid in S and G2 phases of the cell cycle [26,27]. Rad54 protein is a Snf2-related dsDNA-specific ATPase essential for HR mediated by Rad51 [28,29].

To investigate whether RAD54L expression was regulated by E2F1 in cells injured by MMC treatment, 5637 cells were first treated with MMC (10 μM) and the effect of RAD54L expression on the overexpression vector (pE2F1) was analyzed (Figure 4A). In addition, cells with stable knockdown of E2F1 (shE2F1) and control shNTS cells were treated with MMC and the expression of RAD54L was compared (Figure 4B). Overexpression of E2F1 in 5637 cells significantly increased RAD54L expression in the presence of MMC (Figure 4A,C). In cells with E2F1 depletion (shE2F1), the RAD54L expression was significantly reduced compared with that of the control (shNTS) after MMC treatment (Figure 4B,D).

The DNA comet assay was performed to determine whether the regulation of RAD54L by E2F1was functionally involved in DSB repair. MMC was used to induce DNA damage in 5637 cell lines. The degree of DNA repair was measured by the comet tail length in a neutral gel (DSB-detectable) [30]. The 5637 cells were transfected with pE2F1, pRAD54L, or both vectors simultaneously, followed by treatment with 10 μM MMC for 4 h to induce DNA DSBs. The degree of DNA repair was measured for each group following the recovery (Figure 5A). The repair of DSBs generated by MMC treatment showed shorter tails in cells transfected with pE2F1 or pRAD54L than the control group, leading to a high recovery rate of DSBs. In addition, a synergetic recovery effect was observed when both pE2F1 and pRAD54L vectors were transfected simultaneously (Figure 5A,B).

DR-GFP cells were analyzed to measure the HR ratio in living cells (Appendix A) [31,32]. We previously transfected each vector into U2OS DR-GFP cells to determine whether E2F1 and RAD54L played a role in HR repair [32,33]. In the present study, U2OS DR-GFP cells were transfected with control pcDNA, pE2F1, pRAD54L, or bothpE2F1 and pRAD54L simultaneously and then treated with I-sceI enzyme. HR repair activity was then measured as a percentage of GFP-positive cells using FACS (Figure 5C,D). The percentage of GFP-positive cells was the highest in samples transfected with both pE2F1 and pRAD54L (Figure 5B). The percentage of GFP-positive cells in each transfected cell group is plotted in Figure 5D. The results confirmed that RAD54L and E2F1 increased HR recovery when DSB was induced in BC cells.

### 2.4. High Expression Levels of E2F1 and RAD54L Are Correlated with Cancer Progression and Poor Prognosis in Patients with Bladder Cancer

Tissue microarrays (TMAs) were used to determine the role of high expression levels of E2F1 and RAD54L in cancer progression and poor prognosis of bladder cancer patients. As shown in Figure 6A, the protein levels of E2F1 and RAD54L were significantly higher in the tissue samples of patients with T3/T4 (late T stage) than in patients with T1/T2 (early T stage) (Figure 6A). These results showed that the expression levels of these two genes were related to cancer progression.

Next, we performed a hierarchical cluster analysis of 102 bladder cancer patients in GSE13507 (Appendix A). Based on the expression analysis of 1,027 genes, we divided 102 bladder cancer patients into a high RAD54L cluster and a low RAD54L cluster. Depending on the expression of E2F1, these two clusters were redefined into four subclusters (Appendix A). We then performed progression-free survival analysis and estimated the prognostic value with a log-rank test. Patients with high RAD54L expression showed poor prognosis in NMIBC (Figure 6B). Further analysis of subdivided clusters revealed that patients with increased expression of both E2F1 and RAD54L showed poor prognosis (Figure 6).

DNA repair proteins are important for the removal of lesions caused by genotoxic anticancer drugs. They are also important factors contributing to the development of resistance and tumor recurrence [34]. Therefore, we sought to determine the association of bladder cancer recurrence with the expression of E2F1 and RAD54L. Immunohistochemical (IHC) evaluation was used to determine protein expression levels of E2F1 and RAD54L in 17 patients with recurrent bladder cancer and 11 patients with non-recurrent bladder cancer. As shown in Appendix A, the tendency of recurrence was high in the group with high E2F1 expression. However, the expression of RAD54 alone was not significantly associated with recurrence. These results might be limited by the small number of cancer tissue samples, which interfered with the estimation of the risk of cancer recurrence based on the expression of E2F1 and RAD54L.

We also performed recurrence-free survival analysis and estimated the prognostic value with a log-rank test. Patients with high RAD54L expression showed poor prognosis in NMIBC (Appendix A). Further analysis of subdivided clusters revealed that patients with high expression of both E2F1 and RAD54L also showed poor prognosis, although their correlation was not statistically significant (Appendix A).

Taken together, these results demonstrate that the expression of E2F1 and RAD54L is significantly correlated with the progression of bladder cancer, and strongly correlated with its recurrence.

## 3. Discussion

Patients with NMIBC constituting more than 80% of bladder cancer diagnosed have a relatively high 5-year survival rate, although they show progression to MIBC and frequent recurrence [2,35]. The activation of repair mechanisms in cancer cells is one of the factors underlying cancer progression or recurrence [34]. Chemotherapy to prevent cancer progression is most commonly used to treat a variety of cancers. It can kill cancer cells by causing DNA damage [36,37]. Defective DNA repair in cells can lead to the accumulation of mutations known to be important factors in cancer development [34]. Meanwhile, it has been reported that increased activity of DNA repair system may affect cancer progression and anticancer drug resistance [38].

Our previous study [17] has shown that E2F1 gene is involved in bladder cancer progression. It recruits DNA repair elements such as NBS1 and RPA to DSBs and regulate RAD51, BRCA1, and BRCA2 [39,40]. Therefore, in this study, DNA repair genes related to E2F1 associated with bladder cancer progression were investigated. We analyzed the GSE13507 data set to identify DNA repair genes closely related to E2F1. Among them, RAD54L, the most relevant gene, plays a role in HR-related repair of DSBs via double-stranded DNA-dependent ATPase as a cofactor for Rad51 [41]. RAD54L can stabilize Rad51-ssDNA filament [42,43] and promote DNA synthesis by separating Rad51 from heterodimeric DNA [44]. Although research on RAD54L in humans is limited, RAD54 mutations in breast and colon carcinomas and some lymphomas have been reported [45]. In particular, it is unclear whether there is a correlation between RAD54L and E2F1 in bladder cancer cells.

This study showed that E2F1 mediated HRR by directly regulating the expression of HR-related RAD54L. Overexpression of these two genes showed a significant correlation with the progression of bladder cancer and the risk of recurrent bladder cancer. Therefore, RAD54L, a DNA repair gene in addition to E2F1, can be used as a biomarker for bladder cancer prognosis. This gene may also be a new chemotherapy target. Considering that the activity of this DNA repair gene is related to the progression of cancer, it is necessary to evaluate the DNA repair ability of cancer cells when chemotherapy is contemplated for cancer patients.

## 4. Materials and Methods

### 4.1. Cell Lines

Human bladder cancer (BC) cell lines EJ, KU7 and 5637 were purchased from the American Type Culture Collection (ATCC, Manassas, VA, USA) and cultured in DMEM and RPMI1640 supplemented with 10% FBS and 1% penicillin/streptomycin (hereafter referred to as ‘complete medium’) (Capricorn Scientific GmbH, Ebsdorfergrund, Germany). RT4 was purchased from the Korean Cell Line Bank (KCLB, Seoul, Korea) and cultured in DMEM complete medium. T24 was kindly provided by Prof. Cha (Kosin University, Busan, Korea). The EJ-shNTS and EJ-shE2F1 cell lines were described in previous studies [46]. These two cell lines were cultured with DMEM complete medium containing puromycin 8 μg/mL. U2OS-DRGFP cells were cultured in RPMI1640 complete medium. All cell lines were incubated at 37 °C with 5% CO_2_.

### 4.2. Plasmids and Reagents

E2F1 overexpression plasmid (pE2F1) was constructed based on previous studies [47]. RAD54L overexpressing plasmid (pRAD54L) was also prepared similarly. Primer sequences used to prepare plasmid constructs are listed in Appendix A. Transfection was carried out using a jetPRIME reagent of 1:3 according to the manufacturer’s protocol [48]. Gene expression at 24 h post-transfection was normalized to that of corresponding empty vector. Mitomycin C (MMC) and puromycin were dissolved in distilled water and stocked at 1 mM and 5 mg/mL (M4287, P8833; Sigma, Shiga, Japan), respectively.

### 4.3. Luciferase Assay

A luciferase assay was performed as described previously [48]. The promoter sequence from −1191 to +210 was cloned by PCR using human genomic DNA as template to generate pGL3-RAD54L (pro-RAD54L). To examine RAD54L promoter activity, pGL3-B, pro-RAD54L, pE2F1, and pRenilla luciferase (pRL) as control reporter vector were used. Firefly and renilla luciferase activities were measured with a Dual-Luciferase Reporter Assay Kit (E1910, Promega, Madison, WI, USA) and analyzed with a VICTOR3^TM^ Plate Reader (PerkinElmer Inc., Waltham, MA, USA) at a wavelength of 540 mm.

### 4.4. Quantitative Real Time-PCR (qRT-PCR)

Total RNA was isolated from BC cells using RNAiso Plus (9109, Takara Bio lnc., Otsu, Japan) and reverse transcribed into cDNA using PrimeScript^TM^ RT Master Mix (RR036A, Takara Bio lnc.). TB Green^TM^ Premix EX Taq-based qRT-PCR was performed on a CFX96^TM^ real time PCR detection system (Bio-Rad, Hercules, CA, USA) as described previously [48].

### 4.5. Western Blot and Chromatin Immunoprecipitation (ChIP) Assay

Proteins from BC cells were lysed with RIPA buffer containing Complete Easy pack protease inhibitor cocktail tablets (04693116001, Roche, Mannheim, Germany). Protein concentration was determined using BCA Assay (#23227, Thermo Scientific, Rockford, IL, USA [47]) following the manufacturer’s instructions. Antibodies used in western blot were used to target E2F1 (A300-765A, Bethyl Laboratories, Montgomery, TX, USA), RAD54L (ab10705, Abcam), and β-Actin (#4967, Cell Signaling, Danver, MA, USA). Immunoreactivity was detected using a WesternBright^TM^ ECL reagent (K-12045-D50, Advansta, CA, USA) with HRP conjugates. Films were exposed at multiple time points to ensure that images were not saturated (47410 19291, Fuji Medical X-Ray Film, Tokyo, Japan). A ChIP assay was performed according to previous studies [30]. The indicated ChIP-qRT PCR primer sets (Appendix A) were used to amplify DNA fragments by qRT-PCR.

### 4.6. Comet Assay and HR Repair Assay

Cells were treated with MMC. The drug-induced DNA damage was investigated using a comet assay kit (4250-050-K, Trevigen, Gaithersburg, MD, USA) under neutral pH conditions. Cells labeled with SYBR Green I dye (containing kit) were observed under a microscope (Carl Zeiss, Oberkochen, Germany). The comet tail length was measured using an Image J program (NIH, Bethesda, MD, USA) [49].

Human osteosarcoma cell line (U2OS DR-GFP) carrying a GFP-based HR repair reporter was used in this study [32]. DR-GFP plasmids was a kind gift of Maria Jasin (Memorial Sloan-Kettering Cancer Center, New York, NY, USA). A schematic diagram of HR repair assay is presented in Appendix A. Experiments were conducted and GFP-positive cells were counted using FACS analysis based on previous studies [32].

### 4.7. Immunohistochemical (IHC) Analysis

Bladder cancer tissue microarrays (TMA) were acquired from AbCam (Cat. Ab178086, AbCam, Cambridge, UK, 1.1 mm in diameter size, Figure 6A). Non-recurrent, recurrent bladder cancer TMA (2.0 mm in diameter, Appendix A) was obtained from University of Pusan (PSU) according to approved IRB protocol (H-1706-002-007) and confirmed by pathologist. Expression of E2F1 and RAD54L proteins in clinical bladder cancer patient tissues was determined immunohistochemically. Except for the expression of the results ’0’, immunoreactivity score was calculated according to the protocol described in a previous study [30].

### 4.8. Microarray Gene Expression Profiling and Clinical Information

We used a gene expression dataset (GSE13507, *n* = 165) of primary patients with bladder cancer. Among 165 patients, 102 were histopathologically diagnosed with primary NMIBC. The remaining 63 patients had primary MIBC. Clinical data including cancer-specific survival (CSS), progression-free survival (PFS), and recurrence-free survival (RFS) were obtained from Chungbuk National University Hospital. Gene expression datasets (GSE13507) were freely available at NCBI GEO database.

### 4.9. Gene Set Networks and Ontology Analysis

GeneMANIA (http://genemania.org), a web-based database and tool for predicting interactions and functions of gene sets on the basis of multiple networks, was used to analyze DNA repair-related gene interactions. Gene ontology analysis was also performed. A result was regarded as significant when *p* value was less than 0.05 and false discovery rate [FDR] was less than 0.2.

### 4.10. Statistical Analysis

GraphPad Prism 8.0 (Graph Pad Software, San Diego, CA, USA) was used for all statistical analyses. Data are presented as means ± standard deviation (SD). For statistical comparisons, *p* values were determined using Student’s *t*-test. For survival analysis, 5-year cancer-specific survival (CSS), progression-free survival (PFS), and recurrence-free survival (RFS) were calculated using the log-rank test with Kaplan-Meier curves. Survival analysis was performed in R 3.6.1 language environment (http://www.r-project.org).

## Figures and Tables

**Figure 1 ijms-21-09025-f001:**
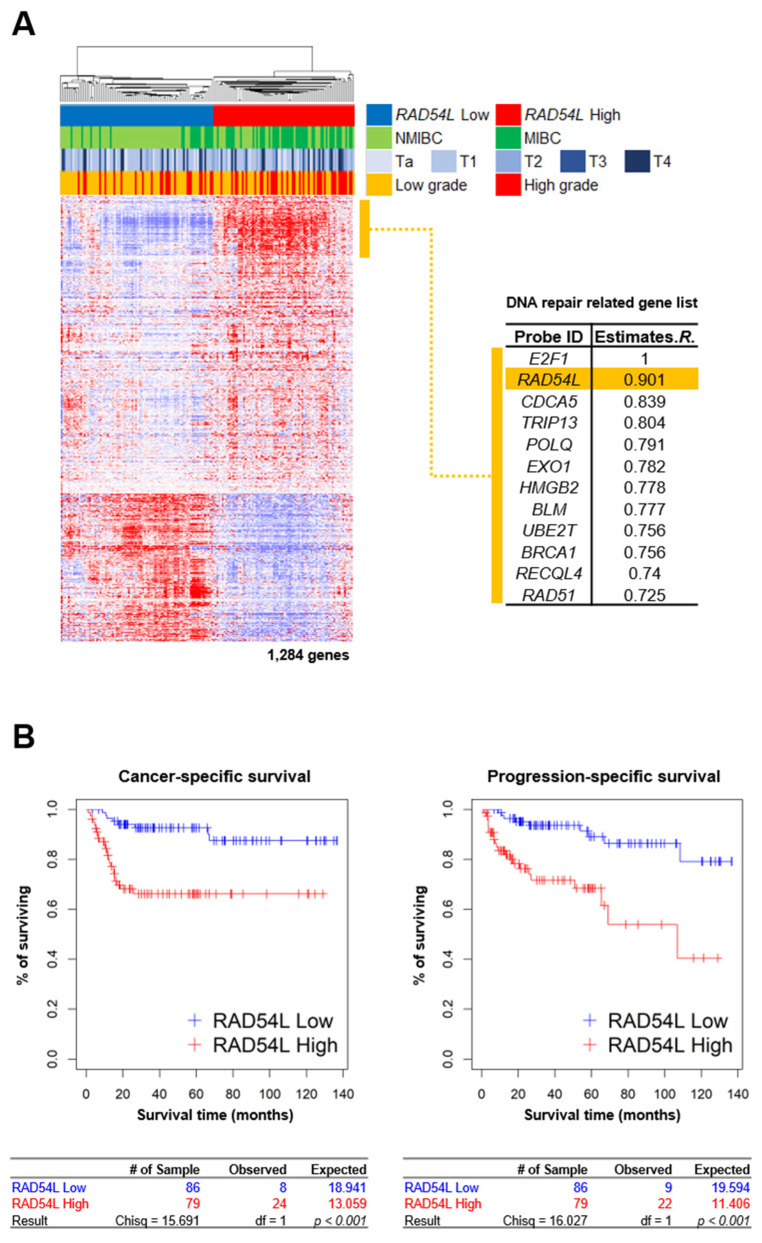
Gene expression patterns of co-expressed gene set of RAD54L, cancer-specific survival, and progression-specific survival of BLC patients in Korean cohort. (**A**) Expression patterns of RAD54L gene and its correlated genes. A total of 1284 genes with expression patterns highly correlated with RAD54L were selected for cluster analysis. (**B**) Cancer-specific survival (CSS) and progression-specific survival (PSS) analyses according to RAD54L expression in BLC patients. BLC patients were divided into two subgroups: low RAD54L expression (Green) and high RAD54L expression (Blue).

**Figure 2 ijms-21-09025-f002:**
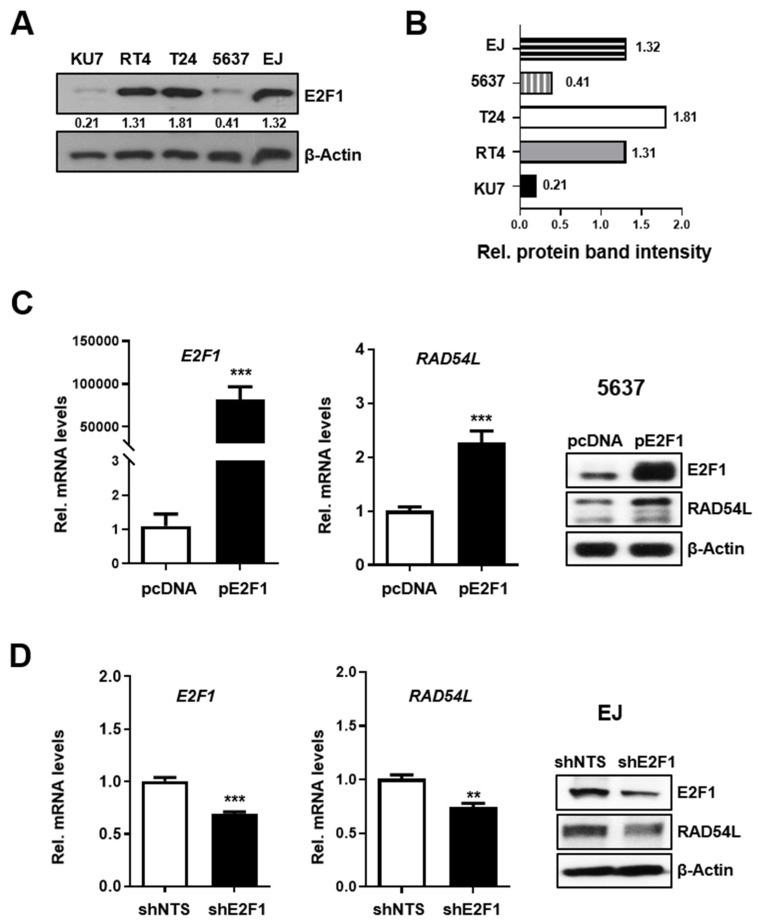
E2F1 regulates RAD54L gene expression. (**A**) E2F1 protein expression levels and (**B**) protein band intensities in BLC cell lines including KU7, RT4, T24, 5637 and EJ cell lines were detected by western blotting analysis. The mRNA and protein levels of E2F1, RAD54L in (**C**) E2F1-transfected 5637 cells, and in (**D**) EJ cells (shE2F1, control cells; shNTS) with a stable E2F1 knockdown were analyzed by qRT-PCR and western blot. Data are presented as mean ± SD. **, *P* < 0.01; ***, *P* < 0.001.

**Figure 3 ijms-21-09025-f003:**
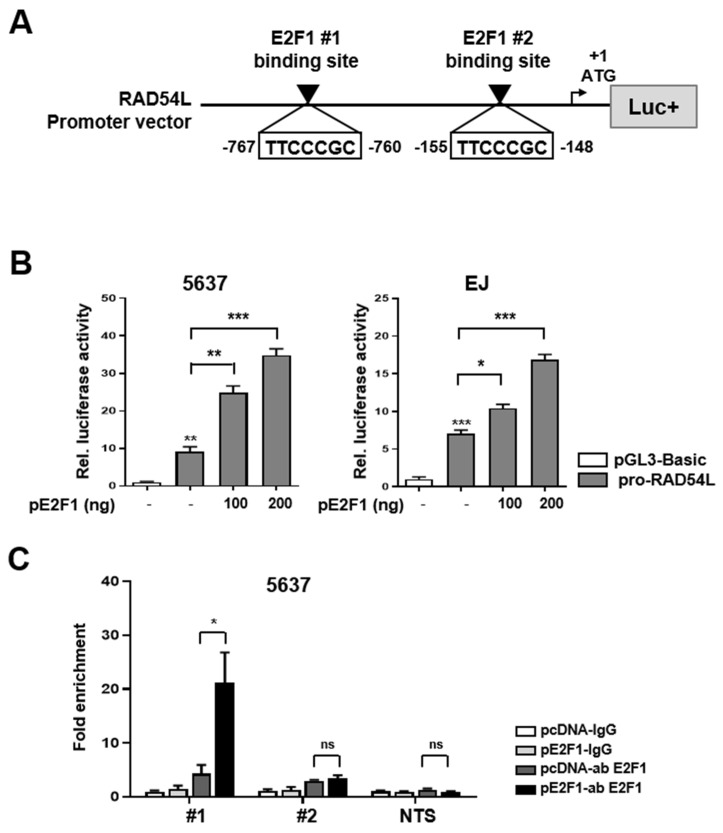
E2F1 is involved in transcriptional activation of RAD54L gene. (**A**) Schematic diagram of RAD54L promoter vector map shown by E2F1 binding sequence. White boxes denote putative E2F1 #1(−767/−760), #2(−155/−148) binding sites. The first ATG is indicated by a black arrow. (**B**) Dual-luciferase reporter assays of RAD54L gene promoter vector were co-transfected with E2F1 overexpression vector (pE2F1) into 5637 and EJ cells. The pGL3-B (pGL3-Basic) was used as the control. Firefly luciferase activity was adjusted with the activity of Renilla luciferase as the internal control. Assays were performed with triplicate samples. *, *P* < 0.05; **, *P* < 0.01; ***, *P* < 0.001. (**C**) ChIP analysis showed recruitment of E2F1 at E2F1-binding sites (#1, #2 and NTS; non-target sequence) on the promoter region of RAD54L. ns, no significant; *, *P* < 0.05.

**Figure 4 ijms-21-09025-f004:**
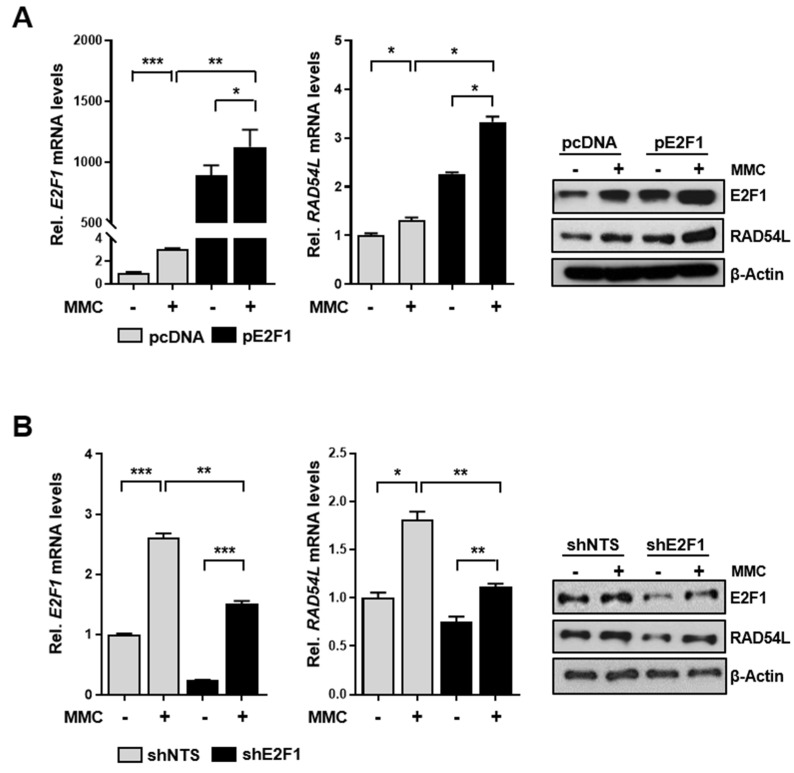
RAD54L expression is induced in MMC-treated cells with E2F1 overexpression or knockdown. (**A**) Cells were incubated without or with 10 μM MMC for 4 h and allowed to recover for up to 24 h. Expression of E2F1 and RAD54L mRNAs and proteins in E2F1 transfected-5637 cells and in (**B**) EJ cells with a stable knockdown of E2F1 (shE2F1, control cells; shNTS) were analyzed by qRT-PCR and western blot, respectively. Data are presented as mean ± SD.*, *P* < 0.05; **, *P* < 0.01; ***, *P* < 0.001.

**Figure 5 ijms-21-09025-f005:**
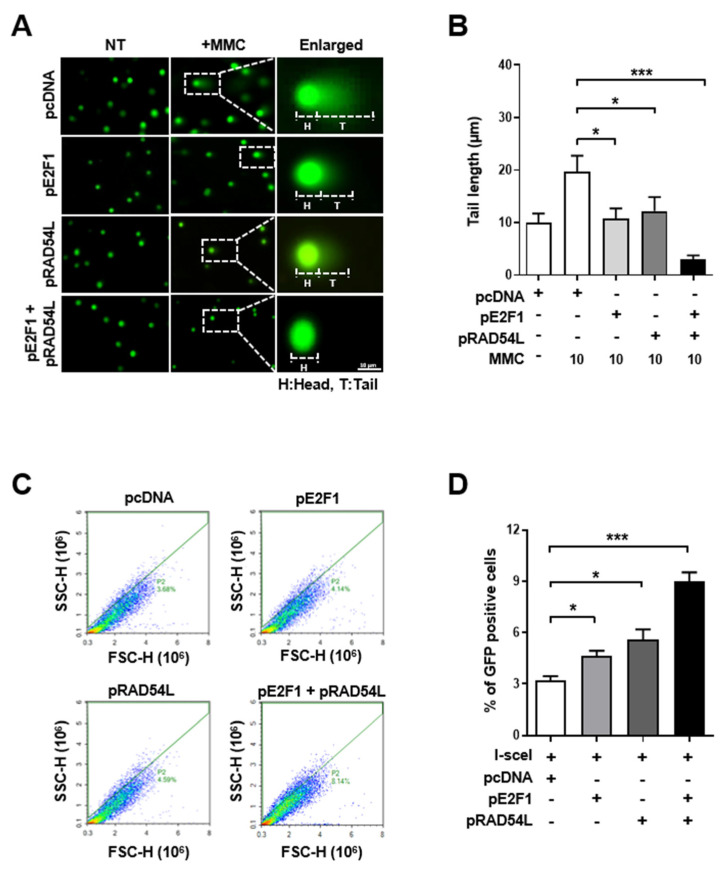
RAD54L and E2F1 increase HR recovery activity when DSB is induced in BC cells. (**A**) Cells transfected with E2F1, RAD54L, or both plasmids were exposed to 10 μM mitomycin C for 4 h before the comet assay was performed. DNA damage was quantified based on tail length. (H, head of comet; T, tail of comet, Scale bar: 10 μm). (**B**) Tail length, (**C**) HR assay by U2OS DR-GFP. (**D**) HR frequency expressed as a percentage of GFP-positive cells in viable whole cell population. Data are presented as mean ± SD. *, *P* < 0.05; ***, *P* < 0.001.

**Figure 6 ijms-21-09025-f006:**
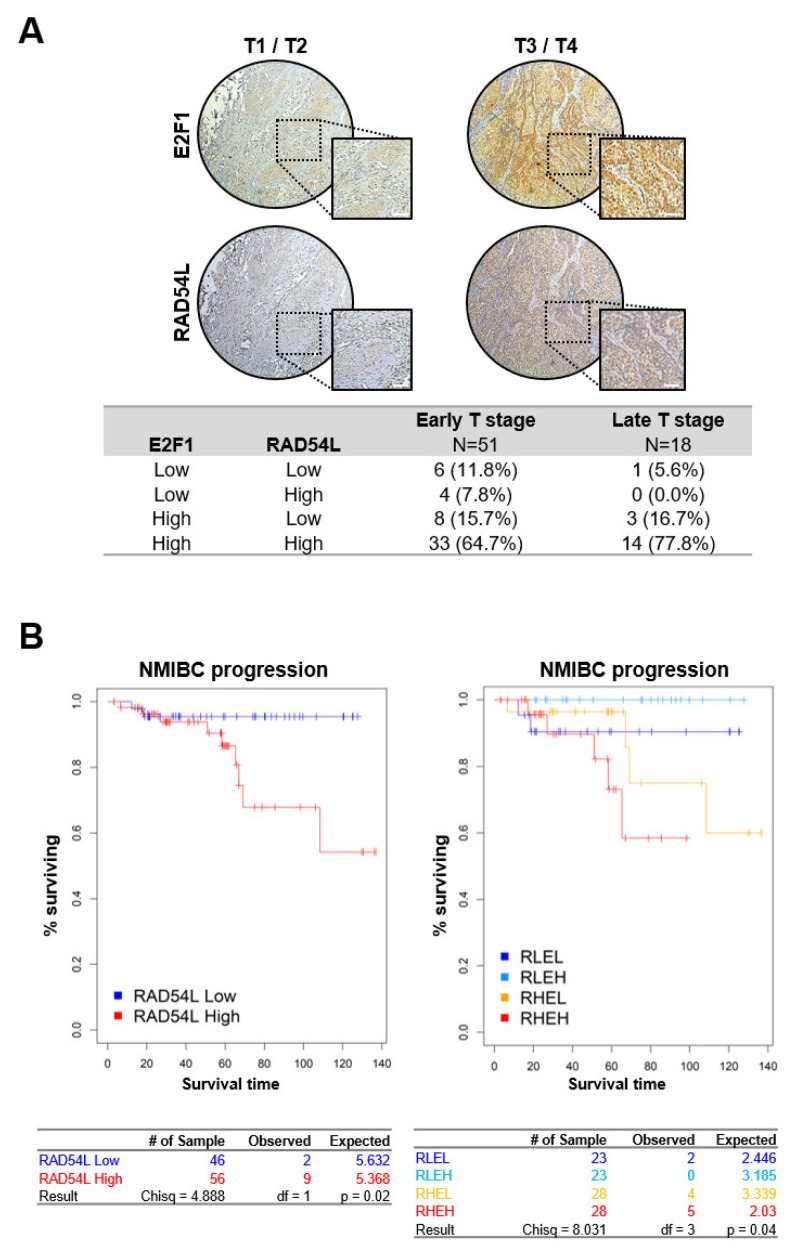
E2F1 and RAD54L expression levels are correlated with cancer progression in BLC patients. (**A**) Immunohistochemical analysis of early and late T stage BLC tissues was performed to detect E2F1 and RAD54L expression in representative patient samples. Scale bar: 100μm. (**B**) Progression-free survival in NMIBC patients (GSE13507, *n* = 102, *P* = 0.02 by log-rank test, left panel). Progression-free survival in NMIBC patients (GSE13507, *n* = 102, *P* = 0.04 by log-rank test, right panel). RLEL, *RAD54L* low and *E2F1* low; RLEH, *RAD54L* low and *E2F1* high; RHEL, *RAD54L* high and *E2F1* low; RHEH, *RAD54L* high and *E2F1* high.

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
