# Peer review of "E2F1 Promotes Progression of Bladder Cancer by Modulating RAD54L Involved in Homologous Recombination Repair"

_ijms, 2020, doi:10.3390/ijms21239025_

Round 1

Reviewer 1 Report

Mun et al have shown nice about E2F1 and 42 RAD54L molecules in bladder cancer. However, the presented hypothesis has been proved in vitro cell data. Do have any animal data to support your finding? Both in vivo and invitro studies could strengthen your hypothesis.

Regarding Fig. 6A, there is enough space available, Could you magnify the image a bit more?  Please indicate the square box position in the circle to show which are you zoom in.

Please mentioned what is that box means which is shown below Fog 6A? Mentioned it in the figure legends too.

Please the article for general grammatical mistakes. 

Author Response

Reviewer #1. (Reviewer Comments to the Author)

  1. Mun et al have shown nice about E2F1 and 42 RAD54L molecules in bladder cancer. However, the presented hypothesis has been proved in vitro cell data. Do have any animal data to support your finding? Both in vivo and in vitro studies could strengthen your hypothesis.

Answer: We agree with the reviewers on the need for in vivo experiments. However, in vivo studies related to E2F1 have already been reported in our laboratory, and because RAD54L acts as an auxiliary molecule in the HR recovery mechanism, it is difficult to derive results with RAD54L alone in vivo experiments as in clinical data. Therefore, this study performed bioinformatics analysis and various cell-based experiments based on clinical data that can confirm the patient's prognosis.

  1. Regarding Fig. 6A, there is enough space available, Could you magnify the image a bit more? Please indicate the square box position in the circle to show which are you zoom in.

Please mentioned what is that box means which is shown below Fog 6A? Mentioned it in the figure legends too.

Answer: According to the reviewer's comment, we showed the mark to the original pictures and added them in the figure legends.

  1. Please the article for general grammatical mistakes.

Answer: This manuscript was revised in English before submission, but grammatical errors in the manuscript were corrected again according to the reviewer's comment.

Reviewer 2 Report

The manuscript “E2F1 promotes progression of bladder cancer by modulating RAD54L involved in homologous 2 recombination repair” reports very interesting finding demonstrating a functional relationship between transcription factor E2F1 and a regulator of homologues recombination RAD54L that contributes to better understanding of molecular mechanisms of bladder cancer progression and suggest these molecules as potential therapeutic targets. The study is well-documented, clearly presented, and can be published in the IJMS.

Please make sure to introduce the abbreviations of NMIBC and MIBS which are used through the paper and not explained (nonmuscle invasive bladder carcinoma and muscle invasive bladder carcinoma).

Author Response

Reviewer #2. (Reviewer Comments to the Author)

The manuscript “E2F1 promotes progression of bladder cancer by modulating RAD54L involved in homologous 2 recombination repair” reports very interesting finding demonstrating a functional relationship between transcription factor E2F1 and a regulator of homologues recombination RAD54L that contributes to better understanding of molecular mechanisms of bladder cancer progression and suggest these molecules as potential therapeutic targets. The study is well-documented, clearly presented, and can be published in the IJMS.

  1. Please make sure to introduce the abbreviations of NMIBC and MIBS which are used through the paper and not explained (nonmuscle invasive bladder carcinoma and muscle invasive bladder carcinoma).

Answer: We really appreciated the reviewer’s comment. We added it.